# Differentiable Weightless Controllers: Learning Logic Circuits for Continuous Control

**Fabian Kresse** [1]   **Christoph H. Lampert** [1]

## Abstract

Controlling autonomous systems under real-world conditions often requires policies that can be evaluated with low latency and minimal energy consumption. Unfortunately, these conditions are at odds with the use of high-precision deep neural networks as controllers. In this work, we introduce Differentiable Weightless Controllers (DWCs), a symbolic-differentiable architecture that learns flexible, non-linear, yet highly efficient control policies. DWCs can be trained end-to-end via gradient-based techniques, yet compile directly into FPGA-compatible circuits with few- or even single-clock-cycle latency and nanojoule-level energy cost per action. Across five MuJoCo benchmarks, including high-dimensional Humanoid, DWCs achieve returns competitive with standard deep policies (full-precision or quantized neural networks). Furthermore, DWCs exhibit structurally sparse and interpretable connectivity patterns, enabling direct inspection of which input values influence control decisions.

## 1. Introduction

Deep Learning has transformed countless fields, from natural language processing (Vaswani et al., 2017; Radford et al., 2019) and computer vision (Krizhevsky et al., 2012; Dosovitskiy et al., 2021) to game playing (Mnih et al., 2013; Silver et al., 2016). The paradigm has been successfully transferred to continuous control as *deep reinforcement learning* (RL), where neural network policies serve as real-valued function approximators, trained to solve highly complex tasks, such as quadrotor racing (Kaufmann et al., 2023), robot locomotion (Miller et al., 2025), and even fusion-plasma control (Degrave et al., 2022).

[1]ISTA (Institute of Science and Technology Austria), 3400 Klosterneuburg, Austria. Correspondence to: Fabian Kresse <fabian.kresse@ist.ac.at>.

*Proceedings of the $43^{rd}$ International Conference on Machine Learning*, Seoul, South Korea. PMLR 306, 2026. Copyright 2026 by the author(s).

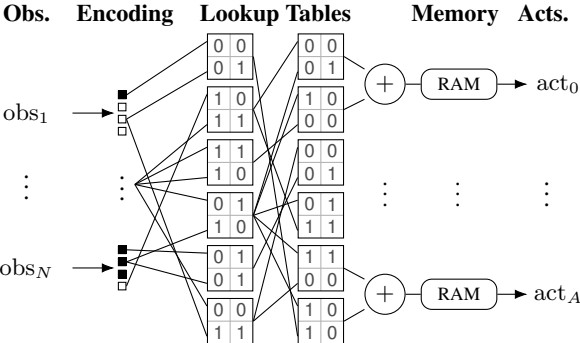

*Figure 1.* Differentiable Weightless Controllers (DWCs): real-valued observations are thermometer-encoded into bitvectors, processed by two layers of multi-input boolean-output lookup tables (here drawn with 2 inputs), aggregated by group summation, and mapped via per-action memory lookups to final action values.

However, standard neural networks rely on large numbers of compute-intense multiply and accumulate operations, which makes them difficult to run efficiently on resource-constrained platforms, such as UAVs and mobile robots. For such cases, policies implemented as small and discrete-valued functions are preferable, which allow efficient implementation on embedded hardware, such as FPGAs.

Even on high-end GPUs, low-precision floating point or integer-only operations have efficiency advantages. Therefore, several prior works have studied either converting standard-trained deep networks into a *quantized* representation (post-training compressions, PTQ), or learning deep networks directly in quantized form (quantization-aware training, QAT), see e.g., Gholami et al. (2021) for a survey. However, most such works target a static supervised or next-word-prediction setting, in which the input and output domains are discrete to start with, and the task involves predicting individual outputs for individual inputs.

The setup of *continuous control*, i.e. learning control policies for cyber-physical systems such as autonomous robots or wearable devices, is more challenging, because input *states* and output *actions* can be continuous-valued instead of categorical, and the controller is meant to be run repeatedly over long durations, such that even small errors in the control signal might accumulate over time.

Consequently, only few methods for quantized policy learning have been proposed so far. Krishnan et al. (2022) discuss PTQ from a perspective of reducing resources during training in RL and, in QAT ablations, find that low-bit quantization often does not harm returns. Ivanov et al. (2025) investigate both QAT and pruning (i.e., reducing the number of weights) jointly, finding that the combination of high sparsity and 8-bit weight quantization does not reduce control performance. Very recently, Kresse & Lampert (2026) showed that controllers can be trained using QAT with neurons that have weights and internal activation values of only 2 or 3 bits; we evaluate our approach against this baseline in the experiment section.

In this work, we challenge the necessity of relying on *weight-based* neural networks, i.e., those that rely on large matrix multiplications during inference, for learning continuous control policies. Instead, we introduce *Differentiable Weightless Controllers (DWCs)*, a symbolic-differentiable architecture for continuous control in which dense matrix multiplication is replaced with sparse boolean logic. Figure 1 illustrates the inference pipeline.

At their core lie *differentiable weightless networks (DWNs)* (Bacellar et al., 2024), an FPGA-compatible and alternative variant of *logic-gate networks (LGNs)* (Petersen et al., 2022) that were originally proposed for high-throughput, low-energy classification tasks. We extend the DWN architecture to the continuous RL domain, enabling synthesis of control policies that are digital circuits instead of real-valued functions. Specifically, we introduce an adaptive input encoding for converting real-valued input signals into binary vectors, and a trainable output decoding that converts binary output vectors into real-valued actions of suitable range and scale. The resulting DWCs are compatible with gradient-based RL, and we demonstrate experimentally that the learned DWCs match standard weight-based neural network baselines (full-precision or quantized) on MuJoCo benchmarks.

At deployment time, DWCs process continuous observations by quantizing them using quantile binning and thermometer encoding, resulting in a fixed-width bitvector representation for each observation dimension. The bitvectors are concatenated and processed by sparsely-connected layers of boolean-output lookup tables (LUTs) (Bacellar et al., 2024). The final layer produces a fixed number of bit outputs per action dimension. These are summed using a *popcount* operation and converted to a final action value using a single-cycle (SRAM) memory lookup.

As a consequence, DWCs are compatible with embedded hardware platforms, especially FPGAs, which explicitly support LUT operations. Here, DWCs can run with few- or even single-cycle latency and minuscule (e.g. Nanojoule-level) energy per operation, as we demonstrate for the case

of an AMD Xilinx Artix-7.

Besides their efficiency, DWCs also offer a potential gain over standard networks in terms of their interpretability, because they consist of sparse, discrete, and symbolic elements instead of dense, continuous, matrix-multiplications in standard networks. For instance, the sparse connectivity in the first layer allows for direct identification of the input dimensions and thresholds utilized by the controller.

**Contributions.** To summarize, **our main contribution is the symbolic-differentiable DWC architecture** that extends previous DWNs from classification to continuous control tasks. We demonstrate that

- **DWCs can be trained successfully using standard RL algorithms**, reaching parity with floating-point policies (full-precision or quantized) in most of our experiments.
- **DWCs are orders of magnitude more efficient than previous architectures**, achieving few- or even single-clock-cycle latency and nanojoule-level energy cost per action when compiled to FPGA hardware.
- **DWCs allow for straightforward interpretation** of some aspects of the policies they implement, specifically which input dimensions matter for the decisions, and what the relevant thresholds are.

## 2. Background

We briefly review deep reinforcement learning for continuous control, and provide background details on DWNs.

### 2.1. Reinforcement Learning (RL)

Reinforcement learning studies how an agent can learn, through trial and error, to maximize *return* (cumulative reward) while interacting with an environment (Sutton & Barto, 2018). There exist various reinforcement learning algorithms that are compatible with our setting of continuous control (continuous actions and observations). In the main body of this work we investigate the performance of DWCs with the state-of-the-art *Soft Actor-Critic (SAC)* method (Haarnoja et al., 2018). Results for *Deep Deterministic Policy Gradient (DDPG)* (Lillicrap et al., 2016) and *Proximal Policy Optimization (PPO)* (Schulman et al., 2017) can be found in Appendix A.

SAC is an off-policy method that keeps a buffer of previous state transitions and taken actions. It updates the parameters of the policy network based on this buffer with soft-Q value estimations from two auxiliary networks (called *critics*). Actions during training are sampled stochastically from a normal distribution, parametrized as $\mathcal{N}(\mu_\theta, \sigma_\theta)$, with $\theta$ being learned parameters. During deployment, the mean action is used deterministically.

## 2.2. Differentiable Weightless Neural Networks

Weightless networks (Aleksander et al., 1984; Ludermir & de Oliveira, 1994) rely on table lookups instead of arithmetic operations (specifically, matrix multiplications): each neuron is a lookup table (LUT) with $k$ binary inputs and a single-bit output. Because each LUT input is connected to precisely one of the preceding layer outputs, this structure results in a sparsely connected computation graph. Signals remain binary throughout the network, enabling multiplication-free inference that maps directly to LUTs available on FPGAs. For the special case of $k = 2$ (two binary inputs per neuron), one recovers *logic-gate networks* of Petersen et al. (2022).

Although traditional weightless networks were hard to train due to their discrete, non-differentiable structure, recently Bacellar et al. (2024) introduced a way to construct efficient surrogate gradients and a learnable interconnect, calling the resulting networks *Differentiable Weightless Networks* (DWNs). Below, we summarize the DWN forward and training mechanisms.

**Thermometer encoding.** DWNs operate on binary signals, so any real-valued observation has to be discretized. For any input $x \in \mathbb{R}$ and predefined thresholds $\tau_1 < \cdots < \tau_B$, define the *thermometer* encoding (Carneiro et al., 2015),

$$E(x) = \big[\mathbb{1}\{x \geq \tau_1\}, \cdots, \mathbb{1}\{x \geq \tau_B\}\big] \in \{0,1\}^B. \quad (1)$$

For $d$-dimensional signals, each dimension is encoded individually, and the results are concatenated, yielding a $B \times d$-dimensional bitvector overall.

**Logic Layers.** A DWN with $L$ layers comprises binary activation maps $b^{(\ell)} \in \{0,1\}^{D_\ell}$, $\ell = 0, \ldots, L$, where $D_0$ is the size of the encoded input, and for $\ell > 0$, $D_\ell$ denotes the number of LUTs in layer $\ell$. Each LUT is a boolean function with arity $k$. For LUT $i$ we form an address $a_i^{(\ell+1)} \in \{0,1\}^k$ by selecting $k$ bits from $b^{(\ell)}$:

$$a_i^{(\ell+1)} = \big(b_{c_1}^{(\ell)}, \ldots, b_{c_k}^{(\ell)}\big). \quad (2)$$

The selection indices $(c_1, \ldots, c_k)$ define the *interconnect*. They are learned via straight-through estimation during training, as described in Bacellar et al. (2024). While Bacellar et al. (2024) only train the first layer's interconnect, we also make the later ones learnable. This is inspired by results in Kresse et al. (2025a), where training accuracy consistently improves if later-layer interconnects are learnable. Each LUT stores a table, $T$, of $2^k$ binary values. The output bit is the addressed entry,

$$b_i^{(\ell+1)} = T_i^{(\ell+1)}\big[\text{addr}(a_i^{(\ell+1)})\big], \quad (3)$$

where $\text{addr}(\cdot)$ maps the $k$-bit vector to its integer index. Concatenating all output bits yields $b^{(\ell+1)}$.

**Group aggregation.** To allow for multi-dimensional outputs, a DWN's final binary features $b^{(L)}$ are partitioned into

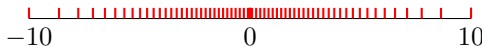

*Figure 2.* DWC thermometer threshold positions.

disjoint groups. For any group $G$, a *group sum* is computed, $s_G = \frac{1}{\tau} \sum_{i \in G} b_i^{(L)}$, with a temperature $\tau > 0$ used as a scale during training. For classification, these $s_G$ serve as logits for a softmax; at inference, the operation reduces to efficient popcount and argmax operations.

**Gradient surrogate.** Because the forward pass is discrete, DWNs rely on surrogate gradients for training. We follow Bacellar et al. (2024), who employ an *extended finite-difference* (EFD) estimator that aggregates contributions from all address locations.

# 3. Differentiable Weightless Controllers

We now provide details on how DWCs extend DWNs to continuous control tasks. In terms of architecture, we adapt the input quantization, such that it can deal with changing input distributions during training, and we adapt the output layer to allow for multi-dimensional continuous actions. Subsequently, we describe the RL training procedure.

## 3.1. Input Encoding

We map observations to binary inputs via clipped normalization and thermometer encoding. For each dimension $j$, we normalize with running mean and standard deviation, subsequently clipping: $\hat{x}_j = \text{clip}\big((x_j - \mu_j)/\sigma_j, -10, 10\big)$. While in deep RL, normalization is often considered optional, we employ it *always*, as we are projecting from $\mathbb{R}$ to a restricted, a priori known interval, for which we can subsequently choose thermometer thresholds $E_j$. After normalization, we employ the same thermometer thresholds for all dimensions $d_{in}$.

For an *odd* number of bits, $B$, we place thresholds at stretched-Gaussian quantiles: let $q_m = m/B$ for $m = 1, \ldots B-1$ and an additional quantile at $\frac{1}{2}$, and set a stretch factor

$$s = \frac{10}{\big|\Phi^{-1}(\frac{1}{B})\big|}, \quad (4)$$

where $\Phi^{-1}$ is the inverse cumulative probability distribution of the standard Gaussian. Define $\tau_{j,m} = s\,\Phi^{-1}(q_m)$ for $m = 1, \ldots, B$ so that the first/last thresholds land exactly at $\pm 10$, and the additional threshold at 0. The thermometer code is $E_j(\hat{x}_j) = [\mathbb{1}\{\hat{x}_j \geq \tau_{j,1}\}, \ldots, \mathbb{1}\{\hat{x}_j \geq \tau_{j,B}\}] \in \{0,1\}^B$, concatenated over $j$ to form $b^{(0)}$ for the first DWN layer. Figure 2 illustrates the resulting input thresholds.

## 3.2. Continuous-control head

To produce continuous actions we reinterpret the group aggregation as a bank of scalar heads, one per action dimension. Let $\{G_1, \ldots, G_{d_{\text{act}}}\}$ be a partition of the final bits $b^{(L)}$. For dimension $d$ we first normalize the group sum:

$$z_d = \frac{s_{G_d}}{|G_d|} - \frac{1}{2} \in \left[-\frac{1}{2}, \frac{1}{2}\right]. \tag{5}$$

We then apply a per-dimension affine transformation:

$$l_d = \alpha_d z_d + \beta_d, \tag{6}$$

with learnable scales $\alpha_d > 0$ and bias $\beta_d \in \mathbb{R}$. To ensure positive $\alpha_d$, we parameterize $\alpha_d = e^{\alpha_{d,p}}$. This head is fully differentiable and integrates with policy-gradient objectives. The emitted $l_d$ is the logit, which is passed through an additional $\tanh$ in the case of SAC before computing the final action. The initialization value of $\alpha_d$ restricts the initial policy actions to a subinterval of the possible action space. This is comparable to initializing the final layer parameters with a low standard deviation, a strategy that has been shown to improve learning in RL (Andrychowicz et al., 2021).

At deployment, the initial threshold operations required for the thermometer encoding should be implemented in a platform-dependent way. Commonly, actual sensor readings are obtained as integers from an *analog-to-digital converter* (ADC), in which the initial bitvector $b^{(0)}$ can be computed as fixed integer-to-thermometer lookup per sensor channel (the constants used for normalization and clipping are fixed after training, so they can be folded into the threshold values).

Subsequently, actions can be computed using only LUT evaluations, popcounts and SRAM memory lookups: starting with $b^{(0)}$, the bitvectors propagate through $L$ LUT layers to produce $b^{(L)}$. For each action head $d$, we popcount its group $G_d$ to obtain the integer $s_{G_d}$. An SRAM then implements the mapping from this popcount to the emitted control word, i.e., $s_{G_d} \mapsto \alpha_d\big(s_{G_d}/|G_d| - \frac{1}{2}\big) + \beta_d$ and, for SAC, the subsequent $\tanh$. In practice, the table would output an integer actuator command.

### 3.3. Training

The learnable components of DWCs are the LUT entries, their connectivity, and the mapping from popcounts to action values. The latter are parameterized as an affine transformation, potentially followed by an additional hyperbolic tangent, so standard gradient-based learning is applicable. To learn the former two in a differentiable way, we rely on the connection learning and the EFD surrogate gradients of Bacellar et al. (2024), making DWCs overall compatible with any gradient-based reinforcement learning algorithm.

Note that, similar to previous work on quantized neural networks (Kresse & Lampert, 2026), during RL training we

*Table 1.* Policy returns in different environments (median and the 25% and 75% quantiles over 10 trained models) for the proposed DWCs, standard floating-point (FP), and low-precision quantized networks (Quant) of Kresse & Lampert (2026). Highest values are marked in bold. DWC returns are comparable to FP and at least as good as Quant across four of the five tasks (all except HalfCheetah).

| Environment | FP | Quant | DWC |
|---|---|---|---|
| Ant | $5.6\text{k}_{[4.3k, 5.8k]}$ | $4.7\text{k}_{[3.9k, 4.9k]}$ | $\mathbf{5.7k}_{[5.5k, 5.9k]}$ |
| HalfCheetah | $\mathbf{11.5k}_{[10.1k, 11.9k]}$ | $10.5\text{k}_{[9.6k, 11.0k]}$ | $7.5\text{k}_{[7.1k, 7.9k]}$ |
| Hopper | $2.8\text{k}_{[2.1k, 3.3k]}$ | $1.9\text{k}_{[1.1k, 3.3k]}$ | $\mathbf{3.1k}_{[2.8k, 3.4k]}$ |
| Humanoid | $\mathbf{6.2k}_{[6.0k, 6.7k]}$ | $6.0\text{k}_{[5.8k, 6.1k]}$ | $6.1\text{k}_{[5.8k, 6.6k]}$ |
| Walker2d | $\mathbf{5.0k}_{[4.7k, 5.2k]}$ | $4.7\text{k}_{[4.4k, 5.0k]}$ | $5.0\text{k}_{[4.5k, 5.2k]}$ |

*only* parametrize the policy networks as DWCs, because only these are required at deployment time. All auxiliary networks, such as the critic networks and the $\sigma_\theta$ head for SAC can remain as standard floating-point networks.

## 4. Experiments

We evaluate the proposed Differentiable Weightless Controllers on five MuJoCo tasks (Todorov et al., 2012): Ant-v4, HalfCheetah-v4, Hopper-v4, Humanoid-v4 and Walker2d-v4, using SAC for training. Our code is publicly available.[1] We also investigate returns for DDPG and PPO in Appendix A.

**Baselines.** As baselines, we use two weight-based neural network setups: First, we use full-precision (FP) models trained with the CleanRL implementation (Huang et al., 2022) of SAC, as reported in Kresse & Lampert (2026). Here, the network has 256 neurons in the single hidden layer. In contrast to the original CleanRL implementation, the networks use running input normalization, as this has been found to improve return for SAC for our tasks. Second, we compare against the QAT-trained low-precision models (2- or 3-bit weights and activations) from Kresse & Lampert (2026). We use their smallest variants that achieve near-FP returns (e.g., Hopper: 16 hidden neurons).

**DWC.** We adopt the same architectures as in the baseline implementations, except that we use DWCs for the policy networks. Unless stated otherwise, DWCs are instantiated with two layers of $k{=}6$ input LUTs. We employ 1024 LUTs per layer, padding the last layer with LUTs to be divisible by the action dimension. Observations are discretized to 63 thermometer thresholds per dimension, as described in Section 3. Ablation studies can be found in Section 5.

**Hyperparameters.** We adopt the protocol used in CleanRL; hence, we use the same hyperparameters across each algorithm for all investigated tasks. For training the DWCs

---

[1]https://github.com/FKresse/
differentiable_weightless_controllers

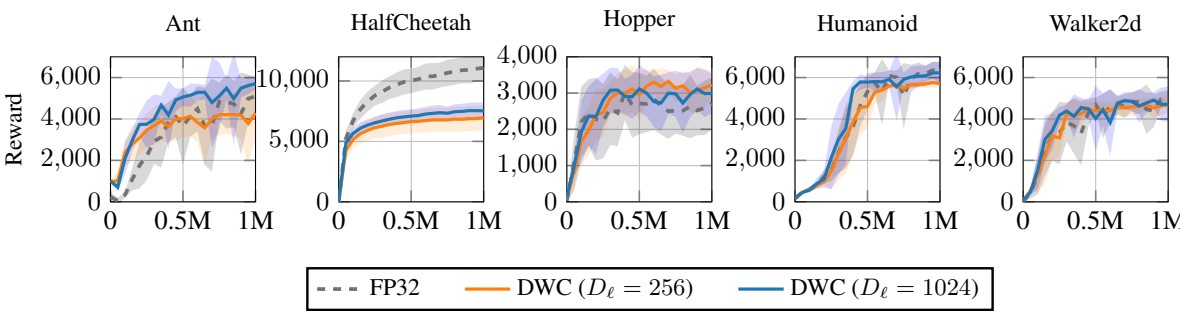

*Figure 3.* Mean return and standard deviation for ten models evaluated across the training steps, ten evaluation episodes per data point and model. Except for HalfCheetah (see main text), training trajectories are comparable to the FP baseline.

| | Environment | Reward | LUTs | FFs | B | DSP | Lat [μs] | P [W] | TP | E.p.A. [J] |
|---|---|---|---|---|---|---|---|---|---|---|
| $D_\ell = 256$ | Ant | $4.5k_{[3.4k, 5.2k]}$ | 0.8k | 0.5k | 0 | 0 | 0.01 | 0.105 | $1.0 \times 10^8$ | $1.1 \times 10^{-9}$ |
| | HalfCheetah | $7.1k_{[6.0k, 7.9k]}$ | 0.8k | 0.4k | 0 | 0 | 0.01 | 0.105 | $1.0 \times 10^8$ | $1.1 \times 10^{-9}$ |
| | Hopper | $\mathbf{3.3k}_{[3.0k, 3.5k]}$ | 0.9k | 0.3k | 0 | 0 | 0.01 | 0.116 | $1.0 \times 10^8$ | $1.2 \times 10^{-9}$ |
| | Humanoid | $5.7k_{[5.5k, 5.8k]}$ | 0.9k | 1.1k | 0 | 0 | 0.01 | 0.102 | $1.0 \times 10^8$ | $1.0 \times 10^{-9}$ |
| | Walker2d | $4.6k_{[4.5k, 4.7k]}$ | 0.8k | 0.4k | 0 | 0 | 0.01 | 0.107 | $1.0 \times 10^8$ | $1.1 \times 10^{-9}$ |
| $D_\ell = 1024$ | Ant | $\mathbf{5.7k}_{[5.5k, 5.9k]}$ | 3.2k | 1.7k | 0 | 0 | 0.02 | 0.225 | $1.0 \times 10^8$ | $2.3 \times 10^{-9}$ |
| | HalfCheetah | $7.5k_{[7.1k, 7.9k]}$ | 3.0k | 2.2k | 0 | 0 | 0.03 | 0.208 | $1.0 \times 10^8$ | $2.1 \times 10^{-9}$ |
| | Hopper | $3.1k_{[2.8k, 3.4k]}$ | 3.2k | 2.0k | 0 | 0 | 0.03 | 0.228 | $1.0 \times 10^8$ | $2.3 \times 10^{-9}$ |
| | Humanoid | $\mathbf{6.1k}_{[5.8k, 6.6k]}$ | 3.2k | 3.7k | 0 | 0 | 0.02 | 0.219 | $1.0 \times 10^8$ | $2.2 \times 10^{-9}$ |
| | Walker2d | $\mathbf{5.0k}_{[4.5k, 5.2k]}$ | 2.8k | 2.1k | 0 | 0 | 0.03 | 0.206 | $1.0 \times 10^8$ | $2.1 \times 10^{-9}$ |
| Kresse & Lampert (2026) | Ant | $4.7k_{[3.9k, 4.9k]}$ | 2.7k | 4.5k | 3 | 45 | 2.29 | 0.39 | $4.4 \times 10^5$ | $8.9 \times 10^{-7}$ |
| | HalfCheetah | $\mathbf{10.5k}_{[9.6k, 11.0k]}$ | 4.3k | 4.6k | 15 | 11 | 243.23 | 0.33 | $4.1 \times 10^3$ | $8.0 \times 10^{-5}$ |
| | Hopper | $1.9k_{[1.1k, 3.3k]}$ | 2.4k | 2.0k | 0 | 45 | 0.21 | 0.31 | $4.8 \times 10^6$ | $6.5 \times 10^{-8}$ |
| | Humanoid | $6.0k_{[5.8k, 6.1k]}$ | 2.3k | 3.1k | 1.5 | 45 | 15.36 | 0.33 | $6.5 \times 10^4$ | $5.1 \times 10^{-6}$ |
| | Walker2d | $4.7k_{[4.4k, 5.0k]}$ | 1.9k | 1.6k | 2 | 4 | 162.23 | 0.17 | $6.2 \times 10^3$ | $2.8 \times 10^{-5}$ |

*Table 2.* Post-synthesis resource utilization for one synthesized model, BRAM (B), end-to-end latency (Lat) in microseconds, estimated power (P) in Watts, peak throughput (TP) in actions per second, and energy per action (E.p.A.) in Joule on an Artix-7 XC7A15T−1 at 100 MHz. Shown reward is the median over all ten models.

we use the same hyperparameters as for the baselines, as specified in Appendix D.

**Training and Evaluation.** For each configuration, we train 10 models with different random seeds. Each model is trained for 1 million environment steps and then its undiscounted return is estimated from 1000 rollouts of the fixed policy from random starting states. Compared to the default CleanRL implementation, we perform all evaluation rollouts with the deterministic, maximum likelihood policy. We investigate the training overhead incured by DWCs, as compared to floating-point networks, in Appendix C.

### 4.1. Results: Quality

Figure 3 shows the training dynamics of DWCs versus the FP baseline. Table 1 shows the resulting returns after training, also including results for quantized networks. Clearly, DWCs learn policies of comparable quality to the floating-point networks, and hence they can readily serve as drop-in replacements, from a reward perspective. An exception is

the HalfCheetah environment, where we observe a substantial return gap. Note that models with a cumulative reward of 7.5k on HalfCheetah are not performing badly at all, indeed they learn to *run*, i.e., master the environment. However, they do so at a slower pace than, e.g., the FP models with 11.5k reward.

Our findings confirm the observation from Kresse & Lampert (2026) that HalfCheetah is the task most resistant to network reduction and quantization, presumably because the task is *capacity-limited*. We study this phenomenon further in Section 5 and the Appendix.

### 4.2. Noise Robustness

Following Duan et al. (2016), we also assess the robustness of DWCs to observation noise. We inject zero-mean Gaussian noise with standard deviations $\sigma \in \{0.1, \dots, 0.5\}$ into normalized observations (unit variance) and compare DWCs to the FP baseline and the quantized noise robustness results from Kresse & Lampert (2026). Figure 4 reports the results,

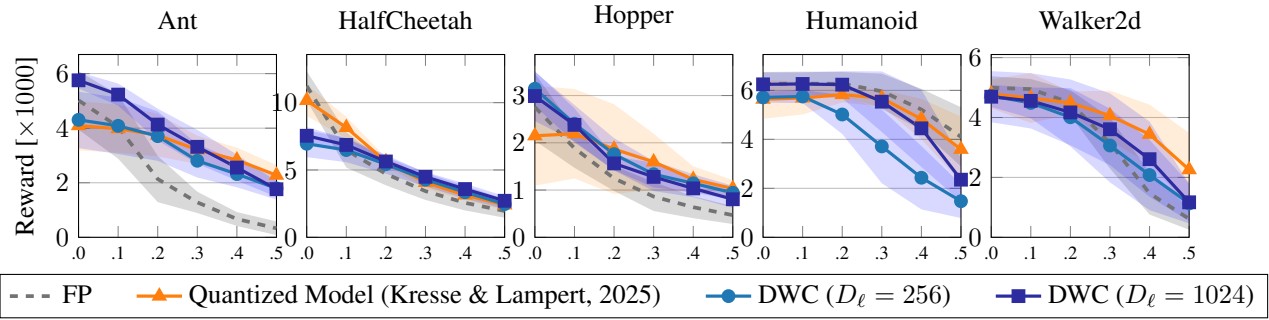

*Figure 4.* Reward performance under injected observation noise with varying noise level $\sigma$. Floating-point (FP), QAT policies from Kresse & Lampert (2026) and our DWCs on MuJoCo tasks. Bands show one standard deviation across trained models. The quantized models and DWCs perform better than, or on par with, the FP baseline under injection, except for Humanoid, where the smaller DWCs show reduced rewards for larger noise.

which show that DWCs achieve noise robustness profiles comparable to that of the quantized network, and generally on the same level as, or better, than floating-point networks. The only exceptions are the compact ($D_\ell = 256$) models, which perform worse on Humanoid under high noise.

### 4.3. Results: Efficiency

The main advantage of DWCs over standard deep neural network controllers is that they consist exclusively of elements that have direct representations on low-energy hardware platforms. To demonstrate this, we perform synthesis and implementation (place-and-route) of the resulting networks for an FPGA, reporting required resources and energy estimates based on the manufacturer-provided tools.

We run out-of-context (OOC) synthesis and implementation with the Vivado toolchain (2022.2), targeting an Artix-7 XC7A15T (speed grade $-1$); see Table 12 in Appendix G for the available device resources. Note that the FPGA only has a total of $10,400$ LUT-6s and twice this amount of flip-flops (FFs), which is a much smaller FPGA than in previous investigations of DWNs (Bacellar et al., 2024). Instead, our setup is directly comparable to Kresse & Lampert (2026), who used the same reference FPGA and OOC synthesis.

We target a clock frequency of $100\,\mathrm{MHz}$, inserting up to two pipeline stages—the first one between the two LUT layers, and the second one before the popcount—until we meet the desired timing. As our design is implemented OOC, we do not consider any overhead due to I/O interfaces. Furthermore, in the main body of this work, we assume that all observation normalization steps and the final per-action lookups take place outside the synthesized core, because these depend on the specific application and interface (e.g. the input format depends on the type of ADCs, the output format on the type of DACs). Note that the quantized baseline (Kresse & Lampert, 2026) likewise does not account for observation normalization. The final per-action mapping can be implemented as an additional single-cycle memory lookup (e.g., in BRAM/ROM), and would therefore add one

cycle to the reported latency. An evaluation including input normalization and SRAM action lookup with representative interfaces is provided in Appendix B.

Table 2 reports reward, LUTs, FFs, BRAMs (B), latency (Lat), power (P), throughput (TP), and energy per action (E.p.A.) for each setup. Since full-precision policies are not practical on the selected hardware, we compare to the quantized networks from Kresse & Lampert (2026). The only difference to their setup is that our power estimates are based on the toolchain's post-implementation power report, rather than the less accurate Vivado Power Estimator.

The results show that DWCs exhibit orders of magnitude lower latency and energy per action compared to the low-bitwidth quantized networks. For our standard $D_\ell = 1024$ setup, latency is only 2 or 3 clock cycles, the throughput is the maximum possible at $10^8$ actions per second, and the energy usage is in the range of 2 nanojoule per action. In contrast, Kresse & Lampert (2026) reports orders of magnitude higher and much more heterogeneous resource usage: their latencies range from 21 cycles to over 24,000 cycles, their throughput between $4.1 \times 10^3$ and $4.8 \times 10^6$ actions per second, and their energy usage per action between $2.8 \times 10^{-5}$ and $6.5 \times 10^{-8}$ Joule. Although control frequency is typically limited by actuation and sensing, a high control throughput can be useful, enabling multiple policy evaluations per step, e.g. for model-based lookahead or safety checks, when paired with a rollout model.

Additionally, in contrast to the quantized models, the computational core of DWCs does not require any BRAM or DSP resources, making them deployable on even more limited hardware than the already small FPGA investigated here.

To illustrate the scaling behavior, we include results for even smaller DWCs with layer width $D_\ell = 256$ in Table 2. This results in single-clock-cycle latency, still maximum throughput, and energy usage per action reduced further by a factor of approximately 2.

# 5. Ablation Studies and Model Interpretability

In this section we report ablation studies on the scaling behavior of DWCs with respect to their layer widths and the input size to the LUTs. Subsequently, we demonstrate how the sparse binary nature of DWCs allows for insights into their learned policies.

## 5.1. Network Capacity

We first investigate the effect of different network capacities. Concretely, we study the effect of varying the layer sizes (widths) and the number of inputs to each LUT. Further ablations on the impact of input resolution and number of layers are available in Appendix F.

**Layer Width.** Assuming that the number of layers is fixed, a large impact on the network capacity comes from the layer width, $D_\ell$, i.e., the number of LUTs. We run experiments in the previous setting with layer width varying across $\{128, 256, 512, 1024, 2048, 4096\}$, where the last layer is padded to be divisible by the action dimension, and report the results in Figure 5. For four of the tasks, the quality is relatively stable with respect to layer widths, with per-layer sizes above 256 generally exhibiting returns similar to the floating-point baseline. This indicates that these learning tasks are not capacity limited, but that at the same time no overfitting effects seem to emerge. As previously observed, HalfCheetah is an exception, where we observe that the quality of the policy increases monotonically with the width of the layers. There are two possible explanations for this phenomenon observed on HalfCheetah: (1) smoother actions, due to a higher output layer width, resulting in more possible actions, as each action head has a resolution of the size of the partition $G_d$; and (2) higher representation capacity and input layer resolution, as more LUTs can connect to different bits in the first layer. In light of the results in Kresse & Lampert (2026) that a 3-bit resolution suffices for HalfCheetah in the output actions; and hidden capacity appears to be the major bottleneck in quantized neural networks, we hypothesize that (2) is the case.

To explore this further, we investigate a substantially larger DWC with $D_\ell = 16,384$ LUTs per layer and inputs quantized to 255 levels (instead of 63). Due to the increased layer width, making the interconnect between the two LUT layers learnable requires $16,384^2 \times k = 1.6\text{B}$ parameters (with $k = 6$), which is prohibitively memory-intensive. We therefore, only for this size, initialize this interconnect randomly and keep it fixed during training. Results are depicted in Figure 5 (dashed light gray entry). With a median return value above $10.3k$, this setup now matches the returns of the quantized baseline (median return $10.4k$) and falls within the region of uncertainty of the floating-point policy (median return $11.5k$). Notably, even at the increased layer width, DWCs require only 32k *lookups* and several popcounts—

substantially fewer than the over 70k multiply-accumulate operations of the two baselines. We take this result as evidence that DWNs can scale to such tasks.

**LUT Size.** Besides the number of LUTs, also the number of inputs per LUT, $k$, impacts the network capacity, as more inputs imply that layers are more densely connected, and that the LUTs themselves can express more complex relations. Figure 9 in the appendix reports average returns for all tasks with $k$ varied from $\{2, 3, 4, 5, 6\}$ and constant layer width $D_\ell = 1024$. In line with the previous experiments, we observe capacity effects limiting returns only for HalfCheetah, while the other returns remain largely unaffected. This suggests that in practice, the number of LUT inputs can be chosen based on what matches the available hardware. For FPGAs, native support for $k = 4$ or $k = 6$ is common, whereas for custom ASICs, smaller values might be preferable (Ahmed & Rose, 2004).

## 5.2. Diagnostics and Interpretability

Deep networks are often criticized as *black boxes* (Molnar, 2020; Vouros, 2022), where the trained policy offers little insight into which feature values drive specific actions.

Here, we demonstrate how binary sparse DWCs can potentially contribute to overcoming this issue to some extent. Because input bits correspond to specific input thresholds and connections to processing LUTs are both sparse and learned, we can infer feature importance simply by counting outgoing connections. Figure 6 illustrates the number of connections received per observation dimension, averaged over the ten trained models ($D_\ell = 1024$). The entries are sorted after aggregation according to the mean number of connections. The data shows that connectivity is non-uniform: some input dimensions receive substantially more connections than others, especially for higher-dimensional observation tasks, i.e. Ant and Humanoid. The low standard deviations across independent models indicate that the network consistently identifies the same specific dimensions as relevant for solving the task.

Interestingly, for Humanoid, a large number of observation dimensions receive no connections at all (on average 275 out of 376 receive a connection). Because the trained model nevertheless attains performance comparable to the FP model, we conclude that the unconnected dimensions are not necessary for good control. In contrast, *torso velocity observations* receive the highest number of connections for Humanoid. As the reward function is heavily dependent on forward velocity, this suggests the DWC has identified the features most directly correlated with the reward signal.

Figure 7 shows the distribution of connections across input threshold bits, averaged over dimensions and over trained models. Recall that the input bit with index 31 out of 63

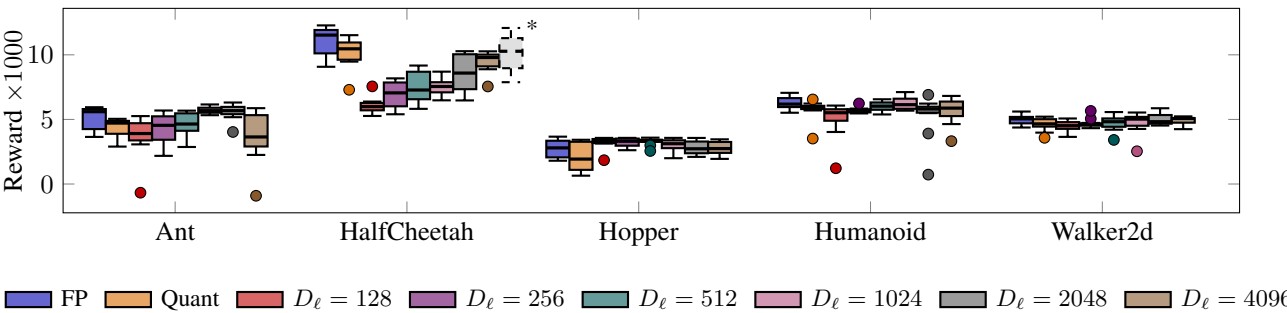

*Figure 5.* Policy returns for FP, Quant and DWCs with varying LUT layer widths. Generally, already models with 256 to 512 LUTs per layer achieve returns on par with the FP baseline. Only for HalfCheetah, we observe a monotonically increasing median return with increasing LUT layer width. * indicates a special high-capacity model with 16k-LUTs per layer and 255-bit per input dimension, see Section 5.

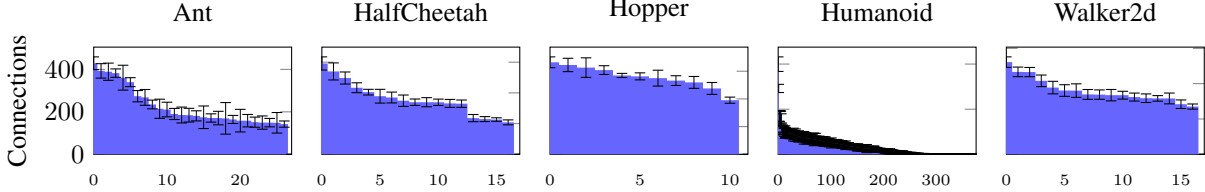

*Figure 6.* Number of connections received for (sorted) input dimension averaged over trained DWCs ($D_\ell = 1024$) for all five environments.

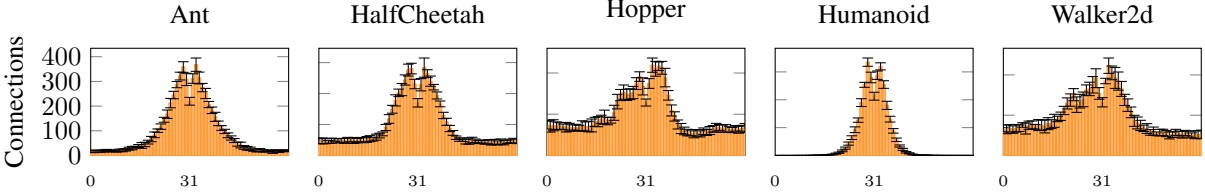

*Figure 7.* Distribution of connections per threshold bit index across all environments. We consistently observe a two-modal distribution slightly left and right of the center (index 31, corresponding to normalized observation value 0).

corresponds to the normalized observation value of 0, index 0 corresponds to the normalized value of $-10$ and index 63 to 10 (Section 3).

Perhaps unsurprisingly, the highest connectivity density is typically found around the normalized observation value of zero. However, instead of a Normal distribution, we observe two distinct modes left and right of the center. This is potentially explained by the reduced probability density captured by the 0 threshold, which is artificially inserted between the two thresholds to its left and right. Furthermore, some tasks (Hopper, Walker, and HalfCheetah) show heavier tails than the others (Ant, Humanoid). We hypothesize that the heavier tails are explainable by the lower observation dimensionality of their respective tasks, and not due to an increased importance of extreme observation values. This hypothesis is supported by Figure 12 in Appendix F, showing the same plot for DWCs with $D_\ell = 128$, which generally exhibit similar return performance (see Figure 5). Here, the tails are lighter, with the maximal number of possible first-layer connections having been significantly reduced to just $k \times D_\ell = 768$, indicating that the heavier tails observed for the larger models are indeed due to

the higher number of possible connections, suggesting that many connections might not be contributing significantly.

## 6. Related Work

**Pruning and Quantization for RL.** Previous work explored quantizing (Krishnan et al., 2022) and pruning deep continuous control RL policies, or both simultaneously (Lu et al., 2024; Ivanov et al., 2025)—which reduces their memory footprint and computational cost— showing that substantial pruning ($\geq 95\%$) (Graesser et al., 2022; Tan et al., 2023) and quantization of most of the network to 3 or 2-bits is possible without harming policy returns (Kresse & Lampert, 2026). Additionally, there has been work on binary, one-bit, quantized RL policies (Valencia et al., 2019; Kadokawa et al., 2021; Chevtchenko & Ludermir, 2021; Lazarus & Kochenderfer, 2022). However, these focus on discrete action spaces or only partially binarize the network. In contrast, our DWCs are fully one-bit policies for continuous control tasks.

**Deep Boolean Networks (DBNs).** Petersen et al. (2022) proposed *Differentiable Logic Networks*, later extending

them to a convolution-style architecture (Petersen et al., 2024b), training layers of 2-input LUTs (Boolean gates) end-to-end with gradient descent.

Since then, several extensions and variants of DBNs have been proposed to address different architectural bottlenecks. Most notably, the original formulation, which required $2^{2^k}$ parameters for $k$-input LUTs, has been adapted to only require $2^k$ parameters per LUT (Bacellar et al., 2024; Ramírez et al., 2025; Gerlach et al., 2025b). Furthermore, interconnect learning was introduced (Bacellar et al., 2024; Yue & Jha, 2024), with extensions to improve training efficiency and similar constructions being proposed (Kresse et al., 2025a; Mommen et al., 2025; Fojcik et al., 2025). Various other adaptations to improve scalability and convergence of DBNs have also been investigated (Kim, 2023; Yousefi et al., 2025; Kim, 2025; Wang et al., 2026).

Most commonly, DBNs have been applied to small-scale image and tabular classification tasks, with some preliminary work on discrete action reinforcement learning based on behavioral cloning (Petersen et al., 2024a). Recently, this has been expanded to include anomaly detection (Gerlach et al., 2025a), recurrent image generation (Miotti et al., 2025), and recurrent language modeling (Bührer et al., 2025). In these applications, scalability has proven to be challenging, due to the high compute requirements during training, as in current work, at least four floating point parameters are required per limited expressivity, Boolean 2-input gate.

Finally, DBNs have shown to be promising for formal verification (Kresse et al., 2025b), an important consideration in control systems, which are often safety-critical.

**Neuro-Symbolic Approaches for RL.** As our work touches upon the intersection of symbolic and neural methods, we very briefly review closely related work in this area. A more comprehensive overview can be found in (Acharya et al., 2023). Bastani et al. (2018) investigate continuous control, distilling policies into decision trees, their approach scales at least to HalfCheetah. Anderson et al. (2020) project policies to a symbolic space. In both approaches, the policy is not directly learned with gradient descent as we do here. Silva et al. (2020) and Kaptein (2025) directly learn a decision tree; however, it is not clear if this approach scales to high-dimensional continuous tasks. In contrast to these works, DWCs directly learn a Boolean structure with gradient descent, which scales to high-dimensional continuous control tasks such as Humanoid.

## 7. Summary and Discussion

In this work, we introduced DWCs (Differentiable Weightless Controllers), which are differentiable weightless networks adapted to handle continuous input states and emit continuous actions. Furthermore, they can be trained using standard gradient-based reinforcement learning algorithms. DWCs allow for highly efficient implementation on low-energy embedded hardware, as we demonstrated by compiling them for an Artix-7 FPGA at 100 MHz, where the resulting networks have extremely low resource requirements at substantially lower inference time (1–3 clock cycle latency) and nanojoule-level energy usage per action. At the same time, they can achieve return parity with standard floating-point networks even on high-dimensional, difficult RL tasks.

We highlighted the interpretability and capacity properties of DWCs through a sequence of experiments and ablation studies, particularly on the HalfCheetah environment, which is the most capacity-limited one of the studied MuJoCo tasks.

An existing limitation is the high computational cost at training time, which significantly exceeds the cost at deployment, because of the relaxations required to allow for gradient-based training. This aspect also means that training is currently only feasible in simulated environments, not interactively on-device.

## Acknowledgments

This work was partially supported by the Austrian Science Fund (FWF) [10.55776/COE12] and the European Research Council (ERC-2020-AdG 101020093, VAMOS). The research was supported by the Scientific Service Units (SSU) of ISTA through resources provided by Scientific Computing (SciComp). Finally, we thank the anonymous reviewers for their constructive feedback, which helped improve the evaluations presented in this manuscript.

## Impact Statement

This work advances machine learning for real-time control by enabling low-latency, energy-efficient policy inference on FPGA hardware. Potential impacts include reduced energy use and improved interpretability for edge deployment.

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

| Alg. | Environment | FP ± sd | DWC ± sd | FP median [$q_{25}$, $q_{75}$] | DWC median [$q_{25}$, $q_{75}$] |
|---|---|---|---|---|---|
| **SAC** | Ant-v4 | 5.1k$_{\pm 0.9k}$ | 5.7k$_{\pm 0.3k}$ | 5.6k$_{[4.3k,5.8k]}$ | 5.7k$_{[5.5k, 5.9k]}$ |
| | HalfCheetah-v4 | 11.1k$_{\pm 1.2k}$ | 7.5k$_{\pm 0.7k}$ | 11.5k$_{[10.1k,11.9k]}$ | 7.5k$_{[7.1k, 7.9k]}$ |
| | Hopper-v4 | 2.7k$_{\pm 0.7k}$ | 3.0k$_{\pm 0.6k}$ | 2.8k$_{[2.1k,3.3k]}$ | 3.1k$_{[2.8k, 3.4k]}$ |
| | Humanoid-v4 | 6.3k$_{\pm 0.5k}$ | 6.2k$_{\pm 0.5k}$ | 6.2k$_{[6.0k,6.7k]}$ | 6.1k$_{[5.8k, 6.6k]}$ |
| | Walker2d-v4 | 5.0k$_{\pm 0.4k}$ | 4.7k$_{\pm 0.9k}$ | 5.0k$_{[4.7k,5.2k]}$ | 5.0k$_{[4.5k, 5.2k]}$ |
| **DDPG** | Ant-v4 | 1.1k$_{\pm 0.6k}$ | 1.3k$_{\pm 0.9k}$ | 1.0k$_{[0.6k,1.4k]}$ | 1.4k$_{[1.0k, 1.6k]}$ |
| | HalfCheetah-v4 | 11.2k$_{\pm 0.6k}$ | 7.4k$_{\pm 0.8k}$ | 11.3k$_{[10.9k,11.4k]}$ | 7.3k$_{[6.7k, 7.9k]}$ |
| | Hopper-v4 | 2.3k$_{\pm 0.8k}$ | 2.0k$_{\pm 0.4k}$ | 2.2k$_{[2.0k,3.0k]}$ | 2.0k$_{[1.8k, 2.2k]}$ |
| | Humanoid-v4 | 1.9k$_{\pm 0.6k}$ | 1.3k$_{\pm 0.2k}$ | 1.7k$_{[1.5k,2.4k]}$ | 1.2k$_{[1.1k, 1.2k]}$ |
| | Walker2d-v4 | 1.7k$_{\pm 0.5k}$ | 2.1k$_{\pm 0.8k}$ | 1.5k$_{[1.4k,1.8k]}$ | 2.3k$_{[1.4k, 2.6k]}$ |
| **PPO** | Ant-v4 | 0.9k$_{\pm 0.4k}$ | 1.6k$_{\pm 0.1k}$ | 0.7k$_{[0.6k,0.9k]}$ | 1.6k$_{[1.5k, 1.7k]}$ |
| | HalfCheetah-v4 | 1.6k$_{\pm 0.6k}$ | 2.1k$_{\pm 0.6k}$ | 1.5k$_{[1.4k,1.6k]}$ | 2.1k$_{[1.6k, 2.5k]}$ |
| | Hopper-v4 | 1.9k$_{\pm 0.6k}$ | 1.7k$_{\pm 0.6k}$ | 2.1k$_{[1.5k,2.4k]}$ | 1.9k$_{[1.5k, 2.0k]}$ |
| | Humanoid-v4 | 0.5k$_{\pm 0.0k}$ | 0.5k$_{\pm 0.0k}$ | 0.5k$_{[0.5k,0.6k]}$ | 0.5k$_{[0.5k, 0.5k]}$ |
| | Walker2d-v4 | 2.2k$_{\pm 1.1k}$ | 1.4k$_{\pm 0.3k}$ | 2.4k$_{[1.3k,2.9k]}$ | 1.3k$_{[1.1k, 1.6k]}$ |

*Table 3.* Return performance for DWCs and FP baseline. Showing mean ± standard deviation and median [25[th] percentile, 75[th] percentile] over 10 trained models. In contrast to SAC, results for DDPG and PPO are more inconsistent, with DWCs sometimes exceeding and sometimes falling short of FP performance.

| | Environment | Reward | LUTs | FFs | B | DSP | Lat [µs] | P [W] | TP | E.p.A. [J] |
|---|---|---|---|---|---|---|---|---|---|---|
| **$b_{obs} = 12\text{-bit}$** | Ant | 4.2k$_{[3.5k, 5.2k]}$ | 3.5k | 0.6k | 4 | 0 | 0.03 | 0.131 | $1.0 \times 10^8$ | $1.3 \times 10^{-9}$ |
| | HalfCheetah | 6.9k$_{[6.0k, 7.9k]}$ | 2.6k | 0.5k | 3 | 0 | 0.03 | 0.125 | $1.0 \times 10^8$ | $1.3 \times 10^{-9}$ |
| | Hopper | 3.2k$_{[2.9k, 3.6k]}$ | 2.3k | 0.4k | 1.5 | 0 | 0.00 | 0.124 | $1.0 \times 10^8$ | $1.2 \times 10^{-9}$ |
| | Humanoid | 5.7k$_{[5.6k, 5.8k]}$ | 6.5k | 2.7k | 8.5 | 0 | 0.02 | 0.164 | $1.0 \times 10^8$ | $1.6 \times 10^{-9}$ |
| | Walker2d | 4.7k$_{[4.5k, 4.7k]}$ | 2.8k | 0.5k | 3 | 0 | 0.00 | 0.125 | $1.0 \times 10^8$ | $1.3 \times 10^{-9}$ |
| **$b_{obs} = 16\text{-bit}$** | Ant | 4.3k$_{[3.5k, 5.3k]}$ | 4.5k | 0.7k | 4 | 0 | 0.03 | 0.137 | $1.0 \times 10^8$ | $1.4 \times 10^{-9}$ |
| | HalfCheetah | 6.9k$_{[6.0k, 7.8k]}$ | 3.2k | 0.5k | 3 | 0 | 0.03 | 0.130 | $1.0 \times 10^8$ | $1.3 \times 10^{-9}$ |
| | Hopper | 3.2k$_{[3.0k, 3.6k]}$ | 2.9k | 0.4k | 1.5 | 0 | 0.03 | 0.129 | $1.0 \times 10^8$ | $1.3 \times 10^{-9}$ |
| | Humanoid | 5.7k$_{[5.5k, 5.8k]}$ | 8.5k | 3.6k | 8.5 | 0 | 0.02 | 0.181 | $1.0 \times 10^8$ | $1.8 \times 10^{-9}$ |
| | Walker2d | 4.7k$_{[4.4k, 4.6k]}$ | 3.5k | 0.5k | 3 | 0 | 0.03 | 0.130 | $1.0 \times 10^8$ | $1.3 \times 10^{-9}$ |

*Table 4.* Post-synthesis resource utilization for 12-bit and 16-bit signed observations, showing BRAM (B), end-to-end latency (Lat) in microseconds, estimated Power (P) in Watts, peak throughput (TP) in actions per second, and energy per action (E.p.A.) in Joule on an Artix-7 XC7A15T−1 at 100 MHz. All models have two layers with $D_\ell = 256$.

# A. DDPG and PPO returns

In this appendix we provide results for training DWCs with DDPG and PPO. Table 3 shows results for all investigated tasks and algorithms. For SAC and DDPG, we use the same hyperparameters for both DWCs and the FP32 baseline. DWCs trained with DDPG use 1024 LUTs per layer. For PPO we perform hyperparameter tuning for both FP32 and DWCs separately, resulting in 256 LUTs per layer for DWCs. Hyperparameters and search details are provided in Appendix D.

For PPO, we directly use the generated logit $l_a$ as the action, whereas for DDPG, we compute the action by applying a tanh as in SAC.

For the DDPG FP32 baseline, we report results with *unnormalized* observations, as this outperforms observation normalization on our tasks (Kresse & Lampert, 2026). All other experiments (SAC, PPO) use observation normalization also for the FP model. While SAC shows very consistent results for FP and DWCs across all tasks, DDPG and PPO results vary, with DWCs sometimes outperforming and sometimes underperforming FP32. Overall, for DDPG the performance gap for HalfCheetah remains the largest, similar to SAC.

|  | Environment | LUTs | FFs | B | DSP | Lat [μs] | P [W] | TP | E.p.A. [J] |
|---|---|---|---|---|---|---|---|---|---|
| FP | HalfCheetah | 7.3k | 9.0k | 3 | 34 | 0.20 | 0.259 | $1.0 \times 10^8$ | $2.6 \times 10^{-9}$ |
|  | Hopper | 4.8k | 5.5k | 1.5 | 22 | 0.20 | 0.191 | $1.0 \times 10^8$ | $1.9 \times 10^{-9}$ |
|  | Walker2d | 7.6k | 9.0k | 3 | 34 | 0.20 | 0.261 | $1.0 \times 10^8$ | $2.6 \times 10^{-9}$ |

*Table 5.* Post-synthesis resource utilization, assuming the unusual setup where quantization to signed integers is performed on-device, and the FPGA is provided with FP32 values. Showing BRAM (B), end-to-end latency (Lat) in microseconds, estimated Power (P) in Watts, peak throughput (TP) in actions per second, and energy per action (E.p.A.) in Joule on an Artix-7 XC7A15T$-1$ at 100 MHz. Two layers with $D_\ell = 256$.

## B. End-to-end synthesis

Here, we provide synthesis results that account for the observation normalization and final affine mapping. Results for $D_\ell = 256$ and two representative quantization bitwidths of the sensors (observations) are given in Table 4. The place-and-route is still performed OOC, and all sensor values are assumed to be already present on the FPGA. We assume sensor bit-widths $b_{obs} = \{12, 16\}$, representing typical values; for instance, see the MPU-6000 IMU commonly used in UAVs (InvenSense Inc., 2013).

As all observation values in the MuJoCo simulator are provided as floating-point values, we chose to perform post-training symmetric quantization of the observation values (without a zero-point). Hence, for $b_{obs}$ bits,

$$Q_{max} = 2^{b_{obs}-1} - 1 \qquad Q_S = \frac{x_{max}}{Q_{max}} \qquad Q(x) = \text{clip}\left( \left\lfloor \frac{x}{Q_S} \right\rceil, -Q_{max}, Q_{max} \right), \tag{7}$$

with $Q_{max}$ being the largest integer representable in the quantized domain, $Q_S$ the quantization scale, and $Q(x)$ the quantization operation. The term $x_{max}$ describes the maximum range of the non-quantized values. To determine it, we performed 10 rollouts with one of our models and set $x_{max}$ to the largest observed absolute value times 1.2, per dimension.

Note that explicit quantization would not be necessary in a real-world setup, as the sensor values are provided as integers to the controller.

The input normalization is implemented by offline computing new thresholds $\tau^*_{i,d}$, with $d$ being the dimension the threshold applies to and $i$ the index within the dimension. These are based on the original thresholds $\tau_{i,d}$, but account for both input normalization and quantization. Then, with $\mu_{running}$ and $\sigma^2_{running}$ representing our frozen normalization statistics,

$$\tau^*_{i,d} = \text{clip}\left( \left\lfloor \frac{\tau_{i,d}\sqrt{\sigma^2_{running,d}} + \mu_{running,d}}{Q_{S,d}} \right\rfloor, -Q_{max}, Q_{max} \right). \tag{8}$$

This folds all multiplication constants into the thermometer thresholds, reducing the normalization and thermometer encoding to integer comparisons. Note that instead of the floor operation in Equation 8, the round-to-nearest operation $\lfloor \cdot \rceil$ could be used for computing the new thresholds. In our implementation, we use the floor operation in Equation 8, and the rewards in Table 4 are reported for this exact choice.

For the output mapping, we insert a single-port BRAM for each action dimension. We perform new evaluation rollouts with the now-quantized observations and recomputed thresholds, explaining minor differences in reward reported in Table 4.

Last, we provide an implementation report in the case where FP32 values are provided to the logic core and quantization handling is included in the reported resources (see Table 5). Note that this is an extremely unusual setup, as sensor values are not provided in FP32; we nevertheless provide these results for completeness. In this setting, we only provide results for HalfCheetah, Hopper, and Walker2d, as the other environments would require time multiplexing the available DSPs or selecting a different FPGA.

## C. Training Overhead

Figure 8 shows training times for different environments for one million training timesteps on an NVIDIA A10 GPU with an AMD EPYC 7513 32-Core Processor. We perform training on a GPU, as the current public implementation of DWNs

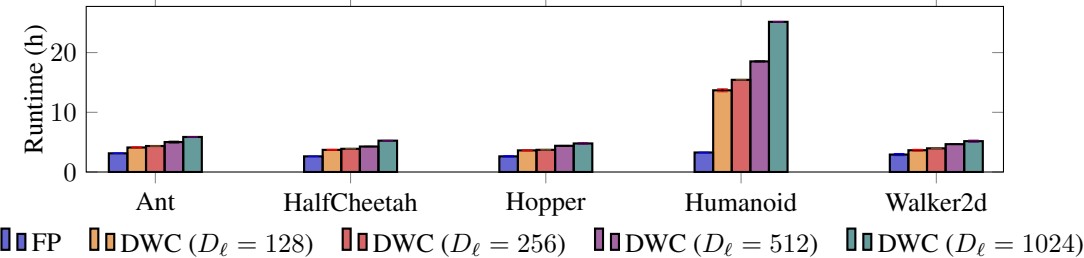

*Figure 8.* Training runtime for one million environment steps, averaged over three training runs (mean $\pm$ std shown; standard deviation is so small that it is not visible). Floating-point models exhibit consistently shorter training times. However, it is only in the Humanoid case, which has a much larger number of observation dimensions than the other environments, that the training time for DWCs is more than $\times 3$ that of the floating-point baseline.

only supports GPU inference and training. DWNs and LGNs are notoriously known for trading high efficiency at inference time (Petersen et al., 2022; Bacellar et al., 2024) for increased training wall clock time. This is especially true when naive trainable interconnects are used (Kresse et al., 2025a), as memory consumption and training time scale with the size of the connectivity matrix. This phenomenon is visible in our results: for almost all tasks, the DWCs required to solve the environments are comparatively modest in size with a small connectivity matrix. Consequently, training times are dominated by the cost of environment simulation steps rather than the network updates, yielding very similar overall times that remain within a $3\times$ factor of the floating-point network's training time up to $D_\ell = 1024$.

The only exception is Humanoid, which exhibits significantly longer training times ($\times 7.7$ when compared to the floating-point baseline for the largest network in Figure 8). The longer training time is due to the much larger interconnect matrix between the first LUT layer and the observations, due to a much larger number of observations.

Future work should consider applying techniques introduced in the classification setting (Kresse et al., 2025a; Mommen et al., 2025; Fojcik et al., 2025; Wang et al., 2026) that reduce the cost of the full-interconnect learning, while retaining most, if not all, performance benefits of the learnable interconnect.

## D. Hyperparameters

Table 6 and Table 7 show SAC and DDPG hyperparameters, respectively. Table 8 shows the PPO floating-point hyperparameters used in our experiments, while Table 9 shows the PPO hyperparameters used for DWCs. Hyperparameters for the LUT and interconnect training with EFD in the DWCs are equivalent to the default configurations in Bacellar et al. (2024).

Major differences between FP and DWCs PPO hyperparameters are a $\times 10$ learning rate for DWCs, a lower entropy coefficient, and a lower max gradient norm. The initial $\log \alpha$ is conceptually similar to the floating-point std parameter, which corresponds to the initialization of the policy network's output variance.

Tables 10 shows the search space for the PPO hyperparameter tuning for DWCs, while Table 11 shows the search space for the PPO hyperparameter tuning for the FP baseline. Remaining hyperparameters were set to the values in Table 8, which correspond to CleanRL (Huang et al., 2022) defaults. Hyperparameter tuning was performed jointly across Ant, Hopper, Walker, and HalfCheetah, optimizing for return after 1M timesteps over 3 seeds. Hyperparameter search was performed with Optuna (Akiba et al., 2019) using the Tree-Structured Parzen Estimator with 100 trials for both the floating-point and DWC implementation. The hyperparameter search for DWCs (PPO) took 25 days of compute, which we distributed over 10 GPUs.

## E. LUT Input Ablation

Figure 9 shows an ablation over the LUT input size $k$ from 2 to 6 inputs. Larger $k$ increases the expressivity of each LUT, but also exponentially increases the number of parameters per layer.

Clearly, already $k = 2$ achieves good results across all tasks, being comparable for all except for HalfCheetah. For HalfCheetah, we observed similar capacity-limited returns as we observed in our size ablation.

*Table 6.* SAC hyperparameters.

| Hyperparameter | Value |
|---|---|
| Total timesteps | 1,000,000 |
| Replay buffer size | $1 \times 10^6$ |
| Discount $\gamma$ | 0.99 |
| Target smoothing $\tau$ | 0.005 |
| Batch size | 256 |
| Learning starts | $5 \times 10^3$ |
| Policy LR | $3 \times 10^{-4}$ |
| Q-network LR | $1 \times 10^{-3}$ |
| Policy update frequency | 2 |
| Target network frequency | 1 |
| Entropy | autotune |
| $\alpha_d$ (DWC only) | 0.5 |

*Table 7.* DDPG hyperparameters.

| Hyperparameter | Value |
|---|---|
| Total timesteps | 1,000,000 |
| Learning rate | $3 \times 10^{-4}$ |
| Replay buffer size | $1 \times 10^6$ |
| Discount $\gamma$ | 0.99 |
| Target smoothing $\tau$ | 0.005 |
| Batch size | 256 |
| Exploration noise (std) | 0.1 |
| Learning starts | $2.5 \times 10^4$ |
| Policy update frequency | 2 |

*Table 8.* PPO FP hyperparameters.

| Hyperparameter | Value |
|---|---|
| Total timesteps | 1,000,000 |
| Learning rate | $7.82 \times 10^{-4}$ |
| Number of environments | 1 |
| Steps per environment | 2048 |
| Number of minibatches | 32 |
| Update epochs | 10 |
| Discount $\gamma$ | 0.99 |
| GAE $\lambda$ | 0.95 |
| Clip coefficient | 0.2 |
| Entropy coefficient | $6.8 \times 10^{-5}$ |
| Value function coefficient | 0.5 |
| Max gradient norm | 0.811 |
| Floating-point std | 0.00212 |
| Adam $\epsilon$ | $1 \times 10^{-5}$ |
| Anneal LR | True |
| Norm advantages | True |
| Clip value loss | True |

*Table 9.* PPO DWCs hyperparameters. Not shown hyperparameters are equivalent to Table 8.

| Hyperparameter | Value |
|---|---|
| Learning rate | $6.76 \times 10^{-3}$ |
| Entropy coefficient | $3.37 \times 10^{-4}$ |
| Max gradient norm | 1.92 |
| Initial log $\alpha$ | -3.18 |
| LUT width | 256 |

*Table 10.* Hyperparameter search space for DWCs.

| Hyperparameter | Search Space |
| --- | --- |
| Learning rate | Log-uniform $[7 \times 10^{-4}, 10^{-2}]$ |
| Max gradient norm | Uniform $[0.5, 2.5]$ |
| Hidden layer size | Categorical $\{128, 256, 512\}$ |
| Entropy coefficient | Uniform $[0, 0.05]$ |
| Initial log $\alpha$ | Uniform $[-6.0, -0.3]$ |

*Table 11.* Hyperparameter search space for FP baseline.

| Hyperparameter | Search Space |
| --- | --- |
| Learning rate | Log-uniform $[7 \times 10^{-4}, 10^{-2}]$ |
| Max gradient norm | Uniform $[0.5, 2.5]$ |
| Entropy coefficient | Uniform $[0, 0.05]$ |
| Floating-point std | Uniform $[10^{-3}, 0.1]$ |

## F. Input Layer Bit & Layer Number Ablation

Figure 10 shows an ablation over the number of thermometer bits used in the input layer. We observe that while performance suffers at 5 input threshold bits, policy performance does not completely collapse. For 63 bits and $l = 128$, Figure 12 shows the learned per-bit index connectivity for all environments. Figure 11 shows an ablation on the number of layers.

## G. XC7A15T FPGA resources

Resources for the AMD Xilinx Artix-7 XC7A15T–FGG484–1 device are shown in Table 12.

*Table 12.* XC7A15T–FGG484–1 device resources.

| Resource | Quantity |
| --- | --- |
| LUTs | 10,400 |
| Flip-flops | 20,800 |
| DSPs | 45 |
| Block RAM (36 Kb) | 25 |
| Max user I/O pins | 250 |

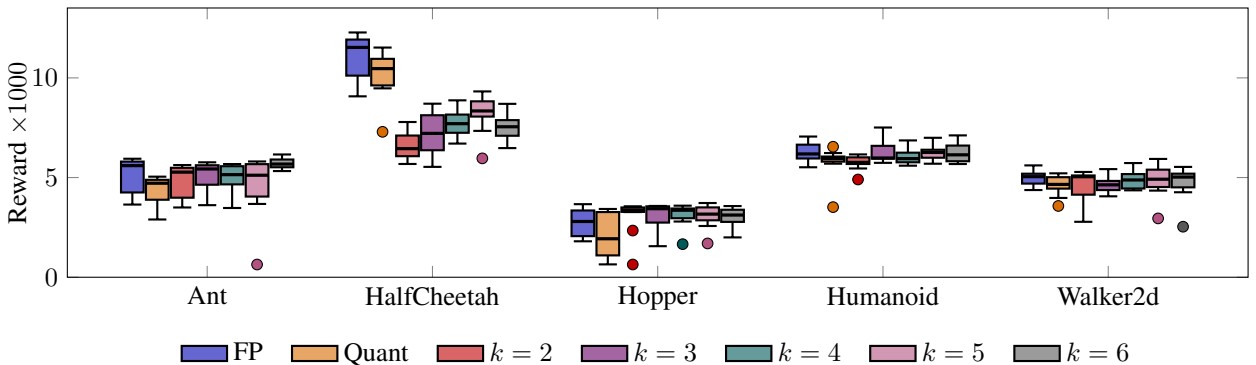

*Figure 9.* Showing floating-point (FP) baseline returns across environments and DWCs with varying LUT-inputs ($D_\ell = 1024$). Generally, already models with 2-input LUTs achieve returns comparable to the FP baseline. Only for HalfCheetah, we observe a generally monotonically increasing median return with increasing LUT table size.

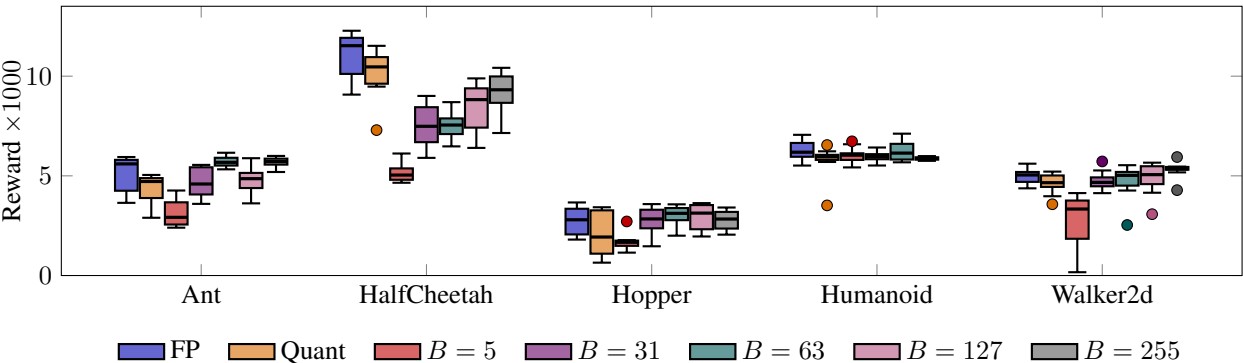

*Figure 10.* Showing floating-point baseline and DWCs return across enviroments with varying input bit width $B$. While having only 5 threshold bits per layer can harm policy performance. For most environments, 63 input bits suffice to saturate returns. As in other cases, the exception is HalfCheetah, which benefits monotonically from more fined-grained input binarizations.

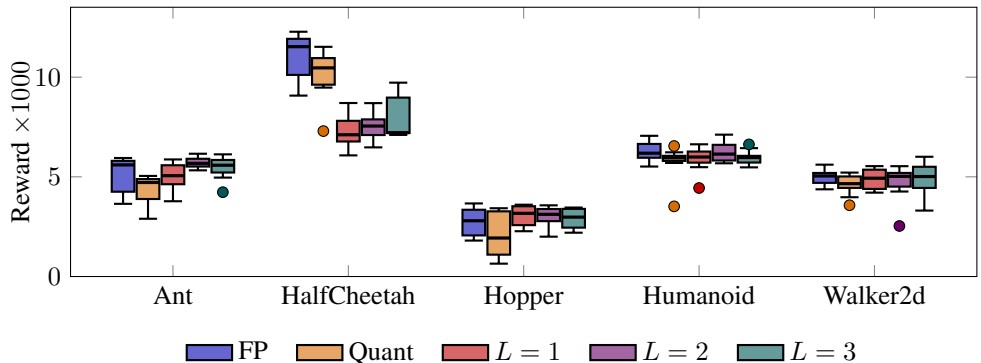

*Figure 11.* Showing floating-point baseline and DWCs return across environments with varying number of layers $L$. For $D_\ell=1024$, one layer generally suffices for attaining floating-point performance. Again, HalfCheetah, seems to benefit from additional layers, and hence, increased capacity.

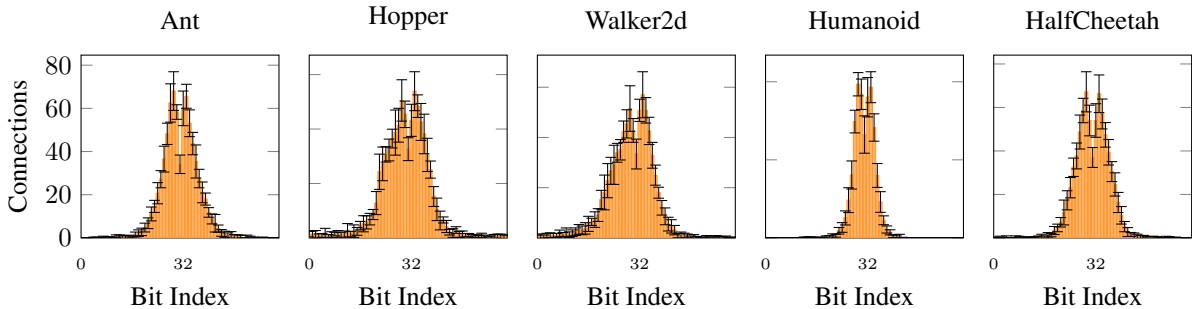

*Figure 12.* Distribution of connections per Bit Index across all environments. $l = 128$

