# OpenReview forum: "Differentiable Weightless Controllers: Learning Logic Circuits for Continuous Control"
_ICML.cc/2026/Conference — ICML 2026 regular_

### Official Review · Reviewer_Go9H · 2026-03-08

**Soundness:** 3
**Presentation:** 3
**Significance:** 4
**Originality:** 2
**Overall Recommendation:** 4
**Confidence:** 4

**Summary:**

The paper studies reinforcement learning controllers designed for efficient hardware deployment under strict latency and energy constraints.

It introduces Differentiable Weightless Controllers, an architecture derived from differentiable weightless networks that replaces multiply–accumulate neural computations with layers of sparse lookup-table logic operating on thermometer-encoded binary inputs. The resulting differentiable logic circuits remain trainable with gradient-based RL methods.

To support continuous control, the authors propose an adaptive binary input encoding and an output head that maps binary features to continuous actions. Controllers are trained with SAC and evaluated on five MuJoCo tasks, achieving performance comparable to floating-point and quantized neural network baselines.

The controllers can be compiled directly into FPGA circuits, yielding extremely low latency

**Compliance With Llm Reviewing Policy:**

Affirmed.

**Final Justification:**

I appreciate the additional context regarding the experimental scope and the discussion of deployment aspects. My main concerns have been adequately addressed, and I am satisfied with the responses.

**Key Questions For Authors:**

The experiments focus on relatively simple MuJoCo tasks. How well does the approach scale to more complex environments, such as tasks with high-dimensional observations (e.g., vision) or more challenging dynamics?

Differentiable weightless networks are often reported to have scaling limitations. Did the authors observe capacity or performance bottlenecks when increasing input dimensionality, LUT size, or network depth?

The hardware evaluation excludes preprocessing steps such as normalization and thresholding. How would these be implemented in deployment, and what impact would they have on latency and energy?

**Limitations:**

Yes

**Strengths And Weaknesses:**

## Strengths

Demonstrates solid empirical results that differentiable weightless networks can be used for continuous control.

Achieves performance comparable to floating-point and quantized neural network policies on most evaluated MuJoCo benchmarks, while at lower energy consumption and latency. Shows extremely low inference latency (≈1–3 clock cycles) and low energy per action through FPGA synthesis results.

Includes multiple experiments, such as robustness tests and capacity scaling that help characterize the behavior of the architecture.

Demonstrates that the architecture maps efficiently to digital hardware.

## Weaknesses

The methodological novelty is limited since the core differentiable weightless network architecture is prior work, and the contribution mainly adapts it to reinforcement learning.

The experiments are limited to relatively simple MuJoCo control tasks. It remains unclear how well the approach scales to more complex environments or higher-dimensional observations. Differentiable weightless networks are currently known to have scalability limitations on more difficult problems.

The computational cost of surrogate-gradient training is mentioned but not quantified.

The hardware evaluation excludes preprocessing steps such as normalization and thresholding, leaving their practical deployment cost unclear.

---

> ### Author Rebuttal · Authors · 2026-03-31
>
> Thank you for your review.
>
> > How well does the approach scale to more complex environments, such as tasks with high-dimensional observations (e.g., vision) or more challenging dynamics?
>
> We have not done such experiments, mostly in order to stay comparable to prior work (Kresse & Lampert, 2025), and because of the high computation load of such tasks. We agree this would be interesting for future work, though.
>
> > Differentiable weightless networks are often reported to have scaling limitations. Did the authors observe capacity or performance bottlenecks when increasing input dimensionality, LUT size, or network depth?
>
> We did not observe scaling issues with training, but indeed, the networks required for MuJoCo are small.
>
> > The hardware evaluation excludes preprocessing steps such as normalization and thresholding. How would these be implemented in deployment, and what impact would they have on latency and energy?
>
> Please see our response to Reviewer nvPw. The impact on latency and energy per action is minor.

---

> > ### Author Rebuttal · Reviewer_Go9H · 2026-04-02
> >
> > Thank you for your clarifications. I appreciate the additional context regarding the scope of your experiments and the impact of preprocessing on hardware deployment. My assessment and recommendation remain the same because the main concerns about the limited methodological novelty and evaluation on relatively simple tasks remain.

---

### Official Review · Reviewer_Ndwn · 2026-03-11

**Soundness:** 3
**Presentation:** 3
**Significance:** 3
**Originality:** 3
**Overall Recommendation:** 5
**Confidence:** 4

**Summary:**

This paper contributes an RL policy architecture, DWC ("differentiable weightless controller"), based on layers of discrete boolean lookup tables, as opposed to continuous neural network weights.  The architecture is an extension of prior work ("differentiable weightless networks") to the context of reinforcement learning, and is well-suited to very high-efficiency FPGA implementation.  The paper tests DWC performance on five standard Mujoco environments for continuous control, in terms of both net reward and various efficiency metrics.  The results also show that DWC performs comparably to, or better than, solid baselines (standard deep policies as well as quantized versions).  Lastly, the paper includes well-directed ablation studies and some feature-attribution results.

**Compliance With Llm Reviewing Policy:**

Affirmed.

**Final Justification:**

Please see my comments in rebuttal acknowledgement; no author response since then.

**Key Questions For Authors:**

- Can the authors comment more on why the affine output map was needed and its impact on performance?  Is there a viable alternative that sticks to LUTs and thermometer encoding/decoding?

**Limitations:**

The authors adequately discuss important limitations in the current work.  In particular, they note the high training cost and consequently the current inability to train on hardware, which could lead to downstream sim-to-real challenges.

**Strengths And Weaknesses:**

Strengths:

- To my knowledge, the paper is technically sound.  The architectural design is well-reasoned, and thorough experimental analyses support all the major claims in the paper.  FPGAs are not my primary area (hence my confidence rating of 4), but in terms of reinforcement learning and control tasks, I did not see any significant technical errors or flaws.

- In particular, I appreciate the well-designed ablation study to get to the bottom of the performance discrepancies in half-cheetah.

- Overall, the presentation is strong; the writing is clear and the figures are high quality.

- The contributions seem quite significant, as computational expense is a major issue in mainstream AI and barrier to deployment on resource-constrained edge devices.

Weaknesses:

- One architectural detail that was under-analyzed was the affine map applied at the final layer, since it breaks the "weightless" paradigm.  I understand that after training this map can be implemented by a lookup, but it left me curious as to why it was included in the architecture.  It would strengthen the paper to analyze this further - for example, removing the affine map in an additional ablation study.

- Originality may be somewhat limited by the fact that DWCs are very similar to the prior DWNs, aside from some small architectural modifications and application to a new task setting (RL).

- A minor presentation issue is that the order of figures and tables does not exactly match their first reference in the main text (e.g. fig 5 before fig 4).

---

> ### Author Rebuttal · Authors · 2026-03-31
>
> Thank you for your encouraging review.
>
> > Can the authors comment more on why the affine output map was needed and its impact on performance? Is there a viable alternative that sticks to LUTs and thermometer encoding/decoding?
>
> If no tanh is employed, a direct popcount-to-action decoding is indeed feasible. However, this would require the popcount range to align directly with the actuator command range.  Therefore, for generality, we included an SRAM transformation in any case.
>
> The affine output map allows the discrete popcount output to be rescaled to a range in which the final tanh can effectively reach saturation when needed. Without it, the output is constrained by the popcount range. The affine map provides a flexible output parameterization, while not incurring inference overhead due to the generally needed SRAM.
>
> > A minor presentation issue is that the order of figures and tables does not exactly match their first reference in the main text (e.g. fig 5 before fig 4).
>
> Thanks for noticing this. We will try to adjust our layout (LaTeX permitting).

---

> > ### Author Rebuttal · Reviewer_Ndwn · 2026-04-01
> >
> > The authors' explanation about remapping to actuator ranges make sense.  As it was a relatively minor point, and there is no new ablation study, I am keeping my score unchanged.

---

### Official Review · Reviewer_nqBX · 2026-03-12

**Soundness:** 3
**Presentation:** 3
**Significance:** 3
**Originality:** 4
**Overall Recommendation:** 5
**Confidence:** 2

**Summary:**

This paper introduces Differentiable Weightless Controllers (DWCs), a novel architecture for continuous-control reinforcement learning (RL) that replaces traditional dense, weight-based neural networks with sparse, boolean lookup-table (LUT) computation. By extending differentiable weightless networks from classification to the continuous domain—using thermometer encoding for inputs and a lightweight continuous-action decoding head—the authors propose a policy representation that is highly efficient for hardware deployment. The method is evaluated on five MuJoCo tasks using Soft Actor-Critic (SAC). Results demonstrate that DWCs achieve performance competitive with floating-point (FP) and quantized baselines on four tasks, though a performance gap remains in HalfCheetah. The paper further provides FPGA synthesis results, showcasing sub-microsecond latency and extremely low energy consumption.

**Compliance With Llm Reviewing Policy:**

Affirmed.

**Final Justification:**

The explanations were helpful for understanding the paper. I am keeping the score.

**Key Questions For Authors:**

- Can the authors clarify if the performance gap in HalfCheetah is primarily due to the resolution of the action space (limited by the number of LUTs in the output group) or the optimization difficulty of the surrogate gradients in high-dimensional tasks?

- Regarding the FPGA implementation, what is the estimated resource overhead and latency of the final affine transformation (decoding) compared to the logic core itself?

- How does the training time (wall-clock) and memory usage of DWCs compare to standard MLP policies during the SAC training loop?

- Since the output is essentially a quantized sum of boolean values, have the authors explored using non-linear decoding or increasing the "popcount" resolution to handle more sensitive tasks?

- You mention folding normalization into thresholds. Does this require re-synthesizing the FPGA bitstream if the running statistics of the environment change significantly during deployment?

**Limitations:**

yes

**Strengths And Weaknesses:**

Strengths:

- The adaptation of weightless/logic-based architectures to continuous control is a significant and non-trivial extension. Handling continuous inputs and outputs through differentiable LUT structures is a fresh direction in the "hardware-friendly RL" literature.

- The FPGA synthesis results are the highlight of the work. The reported latency (0.01-0.03 $\mu s$) and energy efficiency represent orders-of-magnitude improvements over traditional quantized neural networks (QNNs).

- The integration of thermometer encoding, differentiable sparsity, and group aggregation is conceptually sound and well-explained.

- On four out of five MuJoCo benchmarks (Ant, Hopper, Humanoid, Walker2d), the DWCs match or nearly match the performance of full-precision FP32 baselines.

- Unlike many "black-box" models, the paper provides a tangible path toward interpretability by analyzing sparse connectivity and identifying specific input thresholds that trigger certain control responses.

Weaknesses:

- The substantial performance gap on HalfCheetah (approx. 7.5k vs 11.5k for FP) suggests that DWCs may struggle with tasks requiring high-precision coordination or high-capacity state-action mappings.

- The "larger" variant used to improve HalfCheetah results relies on fixed random interconnects rather than learned ones due to memory overhead. This indicates a potential bottleneck where the benefits of differentiable connectivity do not yet scale gracefully to very large logic circuits.

- While the paper mentions that observation normalization can be folded into thresholds, the final affine decoding head (mapping popcounts to actions) still involves memory lookups and arithmetic. A more explicit breakdown of the overhead for the final "decoding" step versus the "logic" core would provide a more complete picture for hardware practitioners.

- The reliance on surrogate gradients and Straight-Through Estimators (STE) can make training more sensitive to hyperparameters compared to standard FP networks. The paper could benefit from a more detailed discussion on the wall-clock time and stability of the training process itself.

- The current analysis focuses on "which" inputs are used. While promising, it doesn't yet bridge the gap to "how" the logic combinations represent complex physical strategies (e.g., gait cycles), which would be the ultimate goal of symbolic interpretability.

---

> ### Author Rebuttal · Authors · 2026-03-31
>
> Thank you for your review.
>
> > Is the resolution of the actions or the surrogate gradient the issue in attaining equality in HalfCheetah?
>
> Thank you for the interesting question. The evidence suggests that the HalfCheetah gap is unlikely to be driven primarily by limited action-space resolution. The layer-width ablation in Figure 4 indicates that HalfCheetah may indeed benefit from higher output resolution. However, because increasing the output resolution also increases overall model capacity, we cannot fully disentangle the two effects. In addition, Kresse & Lampert (2025) show that floating-point-equivalent performance is possible even with 3-bit output quantization. Taken together, these more strongly point to capacity or, perhaps, optimization being the limiting factor.
>
> > What is the overhead for encoding/decoding?
>
> Please see our response to Reviewer nvPw. The impact on latency and energy per action is negligible. In our current implementation, we trade off low latency for higher resource utilization; however, at the cost of additional latency, resource utilization could be substantially reduced.
>
> > How does the training time (wall-clock) and memory usage of DWCs compare to standard MLP policies during the SAC training loop?
>
> Unfortunately, we currently do not have reliable values for training time. Because of the different code bases, we trained the DWCs on GPUs, but the floating-point networks on CPUs. At the same time, current software stacks are highly optimized for MLPs, but not for DWCs, so even on identical hardware, the comparison would not be entirely fair. For Hopper-v4 (on a Nvidia A-10), the SAC training time is approximately twice as long for the DWC with 256 LUTs, at comparable memory, when compared to the floating point baseline on the same GPU.
>
> We will be happy to create a more thorough comparison for a potential camera-ready copy of the manuscript.
>
> > Since the output is essentially a quantized sum of boolean values, have the authors explored using non-linear decoding or increasing the "popcount" resolution to handle more sensitive tasks?
>
> Note that we already employ a non-linear decoding stage: for SAC, the grouped Boolean outputs are summed, then affinely scaled, and finally passed through a tanh, so the final action is not produced by a purely linear decoding of the popcount. Regarding popcount resolution, this is varied indirectly in Figure 4 through the final-layer width: increasing the number of LUTs in the output group increases the effective output resolution. However, this also increases overall model capacity, so we cannot disentangle the effect of output resolution from that of model size.
>
> > You mention folding normalization into thresholds. Does this require re-synthesizing the FPGA bitstream if the running statistics of the environment change significantly during deployment?
>
>
> The described setup resembles a domain adaptation or continual learning scenario.
> We have not experimented in such setups, but indeed, DWCs out of the box are not designed for it, because it determines the statistics only at training time and keeps them fixed for deployment. If a DWC learned this way is deployed in an environment with very different statistics, likely a form of model update/fine-tuning would be required to recover utility (as it would be for ordinary controllers as well). For DWCs as described in the paper, this would require re-synthesizing the FPGA bitstream. In practice, however, we would not suggest such a procedure, but rather see the problem as an interesting direction for future research.

---

> > ### Author Rebuttal · Reviewer_nqBX · 2026-04-04
> >
> > Thank you for the rebuttal. The explanations were helpful for understanding the paper. Please create a more thorough training time comparison in the potential camera-ready manuscript.

---

### Official Review · Reviewer_nvPw · 2026-03-13

**Soundness:** 2
**Presentation:** 2
**Significance:** 3
**Originality:** 3
**Overall Recommendation:** 4
**Confidence:** 3

**Summary:**

The paper introduces a novel symbolic-differentiable architecture, Differentiable Weightless Controllers, designed for high-efficiency, low-latency continuous control on embedded hardware like FPGAs. By replacing traditional arithmetic-heavy matrix multiplications with sparse Boolean logic (Lookup Tables), the authors demonstrate that weightless networks can be trained end-to-end to solve complex MuJoCo benchmarks. The primary contribution lies in the adaptation of input/output encodings that allow logic-gate networks to process continuous states and produce continuous actions.

**Compliance With Llm Reviewing Policy:**

Affirmed.

**Final Justification:**

The rebuttal addessed my concerns.

**Key Questions For Authors:**

I find the technical concept of DWC highly novel and interesting. My current score is a Weak Reject, primarily due to the incomplete hardware evaluation.

I will be happy to raise my score to a Weak Accept if the authors can provide a post-synthesis report that explicitly includes the resource utilization (LUTs, FFs, DSPs) and power consumption for the observation normalization and affine mapping modules on the FPGA. In a real-world scenario where a platform cannot run an FP32 network, these preprocessing steps are mandatory. If you can demonstrate that the DWC framework remains significantly more efficient than quantized baselines (e.g., Kresse & Lampert 2025) even after including these "hidden" costs, the practical value of this work would be firmly established.

**Limitations:**

Yes

**Strengths And Weaknesses:**

Strengths:

1. DWCs achieve nanojoule-level energy consumption and single-digit clock-cycle latency on an Artix-7 FPGA, which is very impressive.

2. The work provides a proof of concept that high-dimensional tasks can be mastered using pure logic circuits rather than weighted neural networks.

Weaknesses:

1. The hyperparameter search for the PPO variant required 25 days of compute on 10 GPUs. This heavy offline training burden stands in stark contrast to the lightweight deployment motivation.

2. The reported hardware efficiency gains exclude the cost of observation normalization and action mapping. Since these modules often require floating-point or high-precision fixed-point arithmetic, their overhead might negate some of the DWC's benefits in a fully embedded implementation.

3. On capacity-limited tasks like HalfCheetah, DWCs show a notable performance gap compared to full-precision baselines, only approaching parity when scaled to a massive 16,384 LUTs. It highlights a trade-off between efficiency and performance.

---

> ### Author Rebuttal · Authors · 2026-03-31
>
> Thank you for your review. We hope our replies address your concerns.
>
> > cost of hyperparameter search for PPO
>
> We report PPO numbers to be compatible with the prior literature. PPO is sensitive to
> hyperparameters. Hence, to report fair results, we performed a thorough hyperparameter search. Note that this is not specific to DWCs. The hyperparameter search for the floating-point networks took 13 days of compute (albeit on CPU).
>
> > post-synthesis report that includes observation normalization & affine mapping
>
> We had not included the input normalization and output mapping as these are sensor-specific, but we would be happy to add typical cases.
>
> For example, here is a new post-synthesis report for $D_l = 256$, which accounts for observation normalization and the affine mapping. Note that we chose a straightforward implementation that increases area, while only marginally increasing latency (+1-2 cycles). The post-synthesis numbers are provided under the assumption that all sensor values are provided as 12-/16-bit, respectively (as is common for ADCs, for instance, see the MPU-6000 IMU commonly used in UAVs).
>
> 12-bit signed observations
> | Environment   | Reward | LUTs | FFs  | BRAM | DSP | Lat [µs] | P [W] | TP | E.p.A. [J] |
> |---------------|--------|------|------|------|-----|----------|-------|----|------------|
> | Ant    	           | 4.2k [3.5k, 5.2k]   | 3528 | 616  | 4    | 0   | 0.03     | 0.131 | $1 × 10^8$ |$ 1.31 × 10^{-9}$        |
> | HalfCheetah | 6.9k [6.0k, 7.9k]    | 2582 | 479  | 3    | 0   | 0.03     | 0.125 | $1 × 10^8$ | $1.25 × 10^{-9}$       |
> | Hopper         | 3.2k [2.9k, 3.6k]    | 2344 | 381  | 1.5  | 0   | 0.03    | 0.124 | $1 × 10^8$  |   $1.24  × 10^{-9}$         |
> | Humanoid    | 5.7k [5.6k, 5.8k]    | 6457 | 2724 | 8.5  | 0   | 0.02    | 0.164 | $1 × 10^8$   |    $1.64× 10^{-9}$         |
> | Walker2d     | 4.7k [4.5k, 4.7k]    | 2784 | 460  | 3    | 0   | 0.03      | 0.125 |  $1 × 10^8$ |    $1.25× 10^{-9}$         |
>
> 16-bit signed observations
> | Environment   | Reward | LUTs | FFs  | BRAM | DSP | Lat [µs] | P [W] | TP | E.p.A. [J] |
> |---------------|--------|------|------|------|-----|----------|-------|----|------------|
> | Ant    	           |4.3k [3.5k, 5.3k]  | 4470| 724 | 4    | 0   | 0.03     | 0.137 |$1 × 10^8$ | $1.37 × 10^{-9}$        |
> | HalfCheetah |6.9k [6.0k, 7.8k]  | 3219| 547 | 3    | 0   | 0.03     | 0.130 | $1 × 10^8$ | $1.3 × 10^{-9}$      |
> | Hopper         |3.2k [3.0k, 3.6k]    | 2891| 425 | 1.5  | 0   | 0.03    | 0.129 | $1 × 10^8$  |  $ 1.29  × 10^{-9}$       |
> | Humanoid    | 5.7k [5.5k, 5.8k]  | 8466| 3619| 8.5  | 0   | 0.02    | 0.181 | $1 × 10^8$   |    $1.81× 10^{-9}$        |
> | Walker2d     | 4.7k [4.4k, 4.6k]   | 3484| 528 | 3    | 0   | 0.03      | 0.130 |  $1 × 10^8$ |    $1.25× 10^{-9}$         |
>
> No additional floating-point operations or, in fact, any multiplications are needed. Since sensor values are obtained as integers, it is sufficient to adapt the thresholding used to generate the temperature encoding so that it accounts for the shift with respect to the mean and the rescaling. This is calculated before synthesis. The affine mapping and tanh are also precomputed and stored in BRAM.  We reran evaluation with simulated integer sensor quantization (the simulator provides floating-point values), explaining the slight reward differences. No retraining was performed.
>
> Even after including normalization and the final output mapping (neglected in Kresse & Lampert (2025)), DWCs exhibit orders of magnitude improvements in latency and energy per action.
>
> The reported version prioritizes low latency over area. The added overhead could be reduced by trading latency for resources, for example, serializing some input comparisons across cycles and time-multiplexing currently underutilized BRAMs. We believe that even the simple, latency-optimized version clearly shows substantial efficiency advantages.
>
> Last, we provide an implementation report in the case where FP32 values are provided to the logic core and quantization handling is included in the reported resources. Note that this is an extremely unusual setup, as sensor values are not provided in FP32; we nevertheless provide it to remove any doubts about the efficiency of the pipeline. Hence, in the table below, we include an additional FP32-quantization stage.
>
> Floating-Point inputs (quantization to 16-bit signed)
> | Environment   | LUTs | FFs  | BRAM | DSP | Lat [µs] | P [W] | TP | E.p.A. [J] |
> |---------------|------|------|------|-----|----------|-------|----|------------|
> | HalfCheetah  | 7293| 8973| 3    | 34 | 0.2 | 0.259 | $1 × 10^8$ | $1.26 × 10^{-9}$ |
> | Hopper            | 4847| 5518| 1.5  | 22 | 0.2| 0.191 | $1 × 10^8$  |  $ 1.19  × 10^{-9}$ |
> | Walker2d       | 7588| 8957 | 3    | 34 | 0.2  | 0.261 |  $1 × 10^8$ |    $1.26× 10^{-9}$ |
>
> The other environments would require time-multiplexing the available DSPs, so in practice, one would probably select a different FPGA.

---

> > ### Author Rebuttal · Reviewer_nvPw · 2026-04-03
> >
> > Thanks for the clarification. I will raise my evaluation accordingly.

---

### Decision · Program_Chairs · 2026-04-30

**Decision:**

Accept (regular)

**Comment:**

This paper addresses a timely problem and received good reviews for demonstrating the practicality of DWC. Some common concerns involve high offline training times, limited conceptual novelty over DWNs, and experiments being limited to simple Mujoco RL tasks. To the extent possible within the camera ready timeline, I encourage the authors to address these concerns as they would further strengthen the contribution.